# Object Pose Estimation Using Edge Images Synthesized from Shape Information

**DOI:** 10.3390/s22249610

**Published:** 2022-12-08

**Authors:** Atsunori Moteki, Hideo Saito

**Affiliations:** 1Graduate School of Science and Technology, Keio University, Yokohama 223-8522, Japan; 2Fujitsu Limited, Kawasaki 211-8588, Japan

**Keywords:** pose estimation, monocular RGB image, edge, ridgeline, deep learning, fine-tuning

## Abstract

This paper presents a method for estimating the six Degrees of Freedom (6DoF) pose of texture-less objects from a monocular image by using edge information. The deep learning-based pose estimation method needs a large dataset containing pairs of an image and ground truth pose of objects. To alleviate the cost of collecting a dataset, we focus on the method using a dataset made by computer graphics (CG). This simulation-based method prepares a thousand images by rendering the computer-aided design (CAD) data of the object and trains a deep-learning model. As an inference stage, a monocular RGB image is entered into the model, and the object’s pose is estimated. The representative simulation-based method, Pose Interpreter Networks, uses silhouette images as the input, thereby enabling common feature (contour) extraction from RGB and CG images. However, estimating rotation parameters is less accurate. To overcome this problem, we propose a method to use edge information extracted from the object’s ridgelines for training the deep learning model. Since edge distribution changes largely according to the pose, the estimation of rotation parameters becomes more robust. Through an experiment with simulation data, we quantitatively proved the accuracy improvement compared to the previous method (error rate decreases at a certain condition are translation 22.9% and rotation: 43.4%). Moreover, through an experiment with physical data, we clarified the issues of this method and proposed an effective solution by fine-tuning (error rate decrease at a certain condition are translation 20.1% and rotation 57.7%).

## 1. Introduction

The research topic of estimating an object’s pose is important from the viewpoint of work support using Augmented Reality (AR) [1] or robot picking [2]. As a target, we refer to a study by Moteki et al. [3]. This study presented a visualization method for inspecting manufacturing defects by estimating the object’s pose and superimposing the object’s CAD model. Though the proposed pose estimation method includes part of the manual manipulation, the fully automatic method is preferable. Our study’s purpose is to estimate the pose of a manufacturing object by using only one image taken of the object. Here, the pose means six Degrees of Freedom (6DoF) parameters, consisting of translation and rotation.

There are numerous kinds of methods to estimate the 6DoF pose of an object from one image [1,2,4,5,6]. In recent years, deep learning (DL)-based methods have been developed and shown remarkable results [7,8,9,10,11,12,13,14]. However, the DL-based methods require a large number of sets of images and pose values as training data. It costs a lot for onsite workers to create these training data. Therefore, we propose a method to train the DL model by using the simulation data; that is, using images rendered by computer graphics (CG) software. In the manufacturing field, the design data of the products are prepared as CAD data in advance. Using simulation data, the ground truth pose is easily acquired, so a training dataset can be created with less cost.

One of the main problems when using simulation data as a training dataset is domain shift [15], where the method trained on simulation data cannot perform well on practical data. In the case of the recent DL-based approach [11,13,14], many methods use an original RGB image as input, so they are also affected by the real environmental conditions such as lighting conditions or surface materials. However, estimating these kinds of parameters and rendering simulation data with the parameters are also difficult problems. Hence, we make use of the common feature extraction from both RGB and CG images to overcome the domain shift. In a previous work, Pose Interpreter Networks [7] use silhouette images as input for pose estimation. Although contours can be obtained commonly from both CG and RGB images, one disadvantage of using contours is inaccurate estimation for rotation.

To solve this problem, we propose a method to make use of edge information extracted from the object’s ridgelines. Industrial products have many straight lines and little texture. Furthermore, the extraction of the ridgelines from a CAD model is not affected by the rendering parameters. Therefore, the proposed method extracts dominant ridgelines from CAD data and creates edge images from the ridgelines for training data. Since this method can take advantage of common features (edge) from both CG and RGB images, the trained DL model can be available for inference.

We evaluated our method by making an original 3D model, which is produced at the factory. In quantitative simulation experiments, we confirmed that edge information is useful for improving the accuracy of pose estimation. Moreover, with this 3D model, we created the original dataset based on the LINEMOD dataset [5]. We evaluated the performance against the domain shift and confirmed effective results through various abbreviation studies.

## 2. Related Work

The object pose estimation methods are roughly divided into two types: depth-based and RGB-based. Depth-based methods [4,6] use point cloud information acquired from a depth camera, such as the Time of Flight (ToF) camera or LiDAR. While these methods use 3D geometric information measured by a depth sensor, they are not applicable to outdoor scenes or large-size objects.

Meanwhile, RGB-based methods are divided into three styles: edge-based, template-based, and DL-based. All these methods calculate some features by textures of the image, but the kind of feature and algorithm differs. As an edge-based method, Han et al. [1] realized edge-based pose estimation of aircraft structural parts by integrating inertial sensor data and the voting scheme. As a template-based method, Konishi et al. [2] proved that ingeniously developed orientation features and data structures make pose estimation fast and accurate. However, these methods are specialized in each target object, and more investigation is needed for processing other objects.

For improving generalization performance, DL-based methods are recently widely proposed. SSD-6D [8] expanded SSD [16] to 3D pose estimation and outperformed depth-based methods in accuracy and speed. SingleShotPose [9] enabled accurate pose estimation by using a Yolo [17]-like architecture to estimate the 2D coordinate of the 3D bounding box which surrounds the object. HybridPose [11] predicted an object’s poses by estimating intermediate representations such as keypoints, edge vectors, and symmetry correspondence.

More recently, an end-to-end approach with various input information marks remarkable performance with public datasets. GDR-Net [12] directly predicted 6DoF pose by using dense correspondence-based intermediate geometric representations. DSC-PoseNet [13] enabled pose estimation with only RGB images and 2D object annotations by dividing the method into two steps, weakly-supervised segmentation, and self-supervised keypoint learning. ZebraPose [14] generated 3D surface code hierarchically in advance, and extracted a multi-layer code by a convolutional neural network (CNN) for estimating the pose. Since these methods make use of texture information from an image at the training stage, fluctuation in the environmental situation greatly deteriorates inference performance.

To prevent the domain shift between training data (CG) and test data (RGB), domain randomization is often used. Tobin et al. [18] trained models on simulated images that transfer to real images by randomizing renderings in the simulator. On the other hand, Sundermeyer et al. [10] developed the Augmented AutoEncoder, which is an improvement of AutoEncoder in terms of parameter deviation. However, domain randomization needs a large amount of data to deal with various real environments.

Another approach to prevent the domain shift is to use common features extracted from both CG and RGB images. Pose Interpreter Networks [7] use silhouette images as input and train CNN models to estimate 6DoF object pose. The network architecture is composed of two cascaded components: Segmentation Network which creates silhouette images from RGB images, and Pose Interpreter Network which estimates pose from silhouette images. The disadvantage of this method is less accurate for estimating rotation because the silhouette image has no feature inside the contour.

Our approach overcomes rotation inaccuracy problem for domain shift by using ridgeline information. Since ridgeline distribution changes more frequently than contour distribution as an object rotates, edge information can represent the rotation of an object. Even in the recent literature [11,12,13,14], ridgeline information is not used. Therefore, we implemented the approach and evaluated the performance.

## 3. Methods

### 3.1. Outline

Figure 1 shows the overview of our method. An overall process is divided into two stages: a training stage and an inference stage. This architecture is adopted in the Pose Interpreter Network [7], whose source code is publicly available. As a training stage, the regression model is trained with training data consisting of sets of edge images and ground truth pose values. First, an edge image is created according to a randomly generated ground truth pose by using the 3D model. In this study, two kinds of edge images are considered: CAD-based edge images and CG-based edge images. The CNN model is trained with the generated dataset. This model outputs 6-DoF pose values; that is, the translation element as a 3D vector p and the rotation element as a quaternion q. As an inference stage, an RGB-based edge image is generated by extracting line segments from an RGB image. Then, the RGB-based edge image is input to the pre-trained CNN and the pose is estimated. The following section describes the details of edge image generation and the process of each stage.

### 3.2. Edge Image Generation

Figure 2 shows the process flow of three kinds of edge image generation. The RGB-based edge images are created from physically captured RGB images, and the line segment detector is adapted for extracting edges. Moreover, CAD-based edge images and CG-based edge images are created by simulation with randomly generated ground truth poses. Since the former is made by projecting the edges of the CAD model, it can represent detailed shape information. On the other hand, since the latter is made by extracting edges from rendering CG images by the line segment detector, it can reproduce the edge distribution of RGB-based images. All these images have a “& mask” option, which adds a mask image to the “only edge” image. Here, an image pixel value of black (mask) is 0 and that of white is 255. This option has the effect of clarifying the object’s contour. The following is the description of how to create these images.

#### 3.2.1. CAD-Based Edge Image

A CAD-based edge image is created by a 3D model of the object. This study uses a 3D model whose format is Standard Triangulated Language (STL, .stl) or Wavefront OBJ (.obj). These formats record information about triangle meshes that consist of the object shape. Our method uses *v*: geometric vertices of each mesh and vn: vertex normals. All lines are judged as being either ridgelines or lines on the plane according to the similarity of the formed angle of normals concerning the adjacent meshes. All extracted ridgelines are connected by the similarity of the angle of the direction vector concerning the adjacent ridgelines. Then, the integrated ridgelines are projected onto the image surface with each ground truth pose. Here, camera parameters, including frame size and focal length, suppose to be obtained in advance.

These projected ridgelines include hidden lines; that is, invisible ridgelines in the current pose’s viewpoint. Hidden lines should be eliminated to reproduce the actual edge appearance. Our method uses a simplified Z-buffer algorithm [19] to judge hidden lines by calculating the depth value of each certain ridgeline and potentially relevant meshes. If the ridgeline is in the back in some meshes, it is regarded as a hidden line. In addition, to be robust against fluctuation of the pose, the averaging filter is processed for each image. Hence, the free parameter to tune for better performance is (1) with or without the mask, (2) the line width of the projected ridgelines, and (3) the kernel size of the averaging filter. We evaluated the performance by changing these parameters in the Results section.

#### 3.2.2. CG-Based Edge Image

A CG-based edge image represents line segments extracted from the rendered image by the CG renderer. As a CG renderer, we use BlenderProc [20], a procedural Blender [21] pipeline for photo realistic rendering. BlenderProc requires the 3D model, the intrinsic and extrinsic camera parameters, and the lighting conditions as input. From the rendered image, line segments are detected by the Line Segment Detector (LSD) [22]. These line segments are drawn onto the image. Like a CAD-based edge image, the averaging filter is processed, if needed. We also evaluated the performance by changing the three parameters mentioned above in the Results section. (Though some rendering parameters or LSD parameters can be also tuned for the performance, we selected only 1 set of parameters for evaluation because there are so many parameters. Automatic parameter estimation by learning method can be an important alternative, and we mentioned it in future work.)

### 3.3. Training

By using CAD-based or CG-based edge images, the CNN model is trained. We made use of the Pose Interpreter Network [7] as a CNN model. This model is ResNet-18 [23] followed by a multilayer perceptron. As the author of Pose Interpreter Networks indicates, the global average pooling layer is also removed from the feature extractor. The multilayer perceptron is composed of one fully connected layer with 256 nodes, followed by two parallel branches corresponding to translation and rotation. Note that we trained only one object class, so the translation branch has three outputs, while the rotation branch has four outputs. As a loss function, $L_{4}$ loss proposed in the Pose Interpreter Network is used. The $L_{4}$ loss showed the best performance in other loss functions. This loss function consists of two terms: the first term means a penalty for easy convergence, and the second term means the sum of 3D distances of sampling points between the ground truth and points transformed by estimated pose. We set the number of sampling points as 1000.

### 3.4. Inference

In the inference stage, an RGB image captured by a camera is used. The resolution of the captured image should be the same as the images used for training. Then, like the CG-based edge image, an RGB-based edge image is created by detecting line segments from the original image. A generated RGB-based edge image is input to the pre-trained CNN model and a pose of the object is estimated.

However, in most cases, the background of the image is cluttered so many irrelevant line segments are detected. Although in this study, we set the region of the object manually, well-known object detection frameworks, such as Yolo [17] are also available. In addition, for creating a mask image, the object’s pose should be known. Although this study used the ground truth pose value from the dataset, semantic segmentation such as Segmentation Network [7] could also be applicable.

## 4. Evaluation

To present the proposed method’s efficacy, we evaluated the pose estimation accuracy using the following procedure. Firstly, the dataset we leveraged is presented. We prepared two kinds of datasets: simulation and physical. Secondly, the experiment methodology is presented. We tested various kinds of conditions to conduct some abbreviation studies. Then, the results of the experiments and considerations are shown.

### 4.1. Dataset

#### 4.1.1. 3D Model

The Oil Change dataset [7] used in Pose Interpreter Networks is not suitable for evaluating the proposed method. The distribution of the ridgelines should be similar to the distribution of the line segments extracted by LSD. However, since the dataset includes various objects shaped by curved surfaces, the proposed method cannot extract enough ridgelines. In other words, the target object of our method should be a convex polyhedron. Therefore, we used the Mixture model [3] shown in Figure 1. This was modeled after the actual manufacturing product, which is a joint part of huge constructions. It is $10\;{\mathrm{cm}}^{3}$ and contains a combination of triangles and spheres. We prepared a 3D model of OBJ format and extracted edge information. Furthermore, 3D point clouds of the Mixture model needed for calculating a loss function were created by Point Cloud Library [24]. There are 1000 point clouds, which is the same as the Oil Change dataset.

#### 4.1.2. Simulation Environment

We prepared CAD-based and CG-based edge images described in Section 3.2. Figure 3 presents an example of these images. We used the same camera intrinsic parameters required for perspective projection as acquired by the physical environment mentioned in Section 4.1.3. As for the random pose, the *blue funnel* object pose information from the Oil Change dataset was applied: that is, 64,000 poses for training and 640 poses for validation. However, by using this predefined pose and the *Mixture* object, out-of-frame or partially occluded images were found. Hence, if a bounding box surrounding the object is not completely included in the frame, the pose was replaced with a re-generated random pose. The parameters for random sampling used in the original Oil Change dataset and the re-generated version are the following:Translation (original) avg.: (0,0,0.6) [m], s.d.: $\left(\sqrt{0.15},\sqrt{0.08},\sqrt{0.2}\right)$;Translation (re-gen.) avg.: (0,0,0.6) [m], s.d.: $\left(\sqrt{0.05},\sqrt{0.03},\sqrt{0.1}\right)$;Rotation (original) avg.: (0,0,0,0), s.d.: (1,1,1,1) (the values were divided by norm);Rotation (re-gen.) the same as original.

#### 4.1.3. Physical Environment

Currently, the public datasets containing the ground truth of the pose exist. However, we could not find the dataset corresponding to the condition of our use case. That is, (1) including meshed CAD data (.stl or .obj) made manually and (2) shaped by several straight ridgelines. For example, LINEMOD [5] includes 13 textureless objects. Since 3D models used in the LINEMOD are constructed based on reconstructed point clouds, there are fewer straight lines. T-LESS [25] contains 30 objects having the following characteristics: (i) textureless, (ii) including similar parts between the objects, and (iii) symmetric architecture. While designed 3D models are included in the dataset, there are fewer objects with straight lines.

Therefore, we created an original dataset compatible with the LINEMOD criteria. Figure 4 shows the details. A physical object is shaped with a 3D printer and painted reddish-brown, similar to the manufacturing product. We placed this object on the center of the board where AR markers are printed. The ground truth object pose is calculated by the ArUco [26] algorithm. Then, we captured images from the viewpoints distributed on the surface of the hemisphere (Figure 4: right). The hemisphere has a varying radius from 45 cm to 95 cm. The total number of captured images is 1788 and the resolution of the image is 320 × 240. Camera intrinsic parameters are calculated using Zhang’s method [27] in advance. The orientation of the camera (Logicool C920) is always focused on the object. We denote it as a focusing pose. The following is the parameter of the pose: (i) longitude: 0–360${}^{\circ }$, (ii) tilt angle: 0–90${}^{\circ }$, (iii) role angle: −45–45${}^{\circ }$, and (iv) distance between an object and a camera: 45–95 cm.

After capturing images of the object, LSD [22] is adapted and line segments are displayed on the image. We used the default value of OpenCV implementation for the LSD parameter. The thickness of displayed line segments is two ways: 1 and 2. We matched the condition of thickness between training and inference images. Figure 5 shows examples of RGB-based edge images.

### 4.2. Methodology

#### 4.2.1. Kinds of Experiments

To present the proposed method’s efficacy, we experimented with conditions shown in Table 1. There are eight types of experiments (E1~E8). In E1~E4, we compared the pros and cons of CAD-based and CG-based edge images. In E5 and E6, models were trained by simulated images, and RGB-based edge images were used as inference data to confirm the performance against the domain shift. In E7 and E8, fine-tuning [28] with RGB-based edge images was adapted. In the fine-tuning process, the model’s weights are modified with a small learning rate. While CAD-based/CG-based and RGB-based edge images both represent the place of the ridgeline, essentially, the tendency of edge distribution differs between both images. Therefore, though it needs additional training data with ground truth labels, the fine-tuning process may affect good performance for pose estimation.

#### 4.2.2. Parameter of Each Experiment

##### Edge Image Generation

As for CAD-based and CG-based edge image generation, we regard the width of the edge, the size of the filter, and with/without mask image as parameters. Figure 6 shows examples of CAD-based edge images. As a previous method (condition S0), we adopted the Pose Interpreter Network, which leverages silhouette images. As the proposed method, we generated 12 kinds of images depending on the width of the edges, the filter size, and *and mask* or not. S1 to S12 denote the indices of the conditions.

To generate CG-based images, the lighting condition should be decided. We used the following parameter at rendering Blender: a kind of source is the point light source, the location of the source is (−6,−5,−5), and the energy of light is 1000. These parameters were chosen so that the image’s appearance resembles a physical capture image.

For fine-tuning E7 and E8, 1788 RGB-based images are divided into 1620 images for training data and 162 images for inference data. Each data contains all ranges of the distance between an object and a camera (45–95 cm).

##### CNN Architecture

The values of the parameters for training, such as learning rate, batch size, and so on, were the same as in the previous method. However, the number of epochs was set to 3000 because all converged data completed training before 3000 epochs. We used a GPU server that has Intel Core i7-6850K CPU (3.60GHz) and NVIDIA Quadro GV100 GPU. It takes approximately 12 h to train the model with 64,000 images.

For fine-tuning of E7/E8 experiment, at first, a training process, which is the same as E1/E3, was performed until 3000 epochs, and then an additional training process that uses 1620 RGB-based images was performed until 1000 epochs.

##### Evaluation Metrics

Evaluation metrics are the error of translation $e_{t}$ (cm) and that of rotation $e_{r}$ (deg). These are the same as Pose Interpreter Networks. *N* means the number of evaluation data, namely, 640 for validation of random pose and 1788 for inference of focusing pose.

### 4.3. Result

The quantitative results are summarized in Table 2 and Table 3. The accuracy of the proposed method outperforms that of the previous method in some conditions at experiments E1~E4 when the edge extraction method matches. For example, at the E1/S10 condition, the error rate decreases by 22.9% in translation and 43.4% in rotation. In particular, rotation error is improved more than translation error. In contrast, when there is a difference in the data source (E5/E6), the accuracy of the previous method is better than that of the proposed method. However, when fine-tuning is adapted (E7/E8), the accuracy of the proposed method is better than that of the previous method. For instance, at the E7/S10 condition, the error rate decreases by 20.1% in rotation and 57.7% at translation.

With respect to the mask’s effect, only edge (S1~S6) images could seldom converge. Especially, in the case of *only edge* images with some filtering, the training is not converged (denoted as N/A). This tendency is found in both CAD-based and CG-based patterns. As for *edge and mask* images (S7~S12), S10 (no filter, width 2) shows comparably the highest performance of all. In the condition of width 1, the edges are sometimes too thin to make the image distinguishable. Furthermore, in the condition of filtering, the effect on the performance is limited. On the whole, however, there is no common tendency in terms of edge width or smoothing parameters. In other words, the condition of the highest performance differs depending on the situation. It means that the parameter of edge width or filter size may be trainable depending on the model shape.

Examples of qualitative results are presented in Figure 7. The S0 row represents the result of the previous method (*only mask*) and the other rows show the result of the proposed method. We chose the result of condition S10 (*edge and mask*, no filter, width 2) because it shows the comparatively highest performance in other conditions. At E5/E6, due to the domain shift, pose estimation is worse than that of E2/E4. In contrast, at E7/E8, pose estimation performance is as high as that of E2/E4 or the previous method. With regard to the distance between an object and a camera, the estimation error is not changed, depending on the distance. That is because both learning and inference datasets include near and far data.

### 4.4. Discussion

#### 4.4.1. Simulation Environment

In the simulation environment, with the appropriate parameter setting, both CAD-based and CG-based methods outperformed the previous method. This indicates that the projected lines have discriminative characteristics for pose estimation. Speaking of the difference between CAD-based and CG-based, CAD-based results are slightly more accurate than CG-based. The reason is that adequate environmental parameters conforming to the physical situation should be designated for CG-based rendering. However, these results are only the case with the *Mixture* model. We will need to evaluate other 3D models that have a similar shape as the actual manufacturing product.

The use of *edge and mask* images is preferable for effective convergence of training. According to experiments E1 and E3, all cases using *edge and mask* images converged adequately. The result suggests that the mask (contour) complements the lack of edge lines for misdetection. Ridgelines and contours both seem to be necessary features for pose estimation tasks with the proposed network.

#### 4.4.2. Physical Environment

The results of E5 and E6 demonstrate that the reason for the performance decrease is derived from the difference in edge extraction tendency between the simulation and physical environment. Figure 8 shows an example of the difference between RGB-based (physical) and CAD-based (simulation). The result of LSD is affected by the lighting condition or parameters for LSD. Hence, false positive or false negative edge misdetection occurs for RGB-based images. On the other hand, because we originally implemented the algorithm for hidden line removal in order to fasten the processing speed, some insufficient edge detection also occurs. By contrast, regarding E6, lighting conditions for rendering CG images should be decided so that the renderer can reproduce the physical environment as close as possible. However, sensing the physical environment is a very costly procedure.

To improve responsiveness to the domain shift, both CAD/CG-based images and RGB-based images should be similar representations. For example, data augmentation that eliminates the part of the edge information seems to resolve the problem. Moreover, these days, learning-based line segment detection algorithms are being developed. For instance, M-LSD [29] is a fast line segment detector. By using this technique, the line detection models may be trained so that they output the edge image, such as CAD-based or CG-based images.

#### 4.4.3. Fine-Tuning

Another effective way to alleviate the domain shift is fine-tuning. As pertains to E7 and E8, almost all *edge and mask* results of the proposed method outperform that of the baseline method. Figure 9 presents examples of learning curves by Tensorboard [30]. This plot shows the change of $L_{4}$ loss in proportion to the progress of training. The CAD-based or CG-based edge images are used for training until epoch 3000, and RGB images are used until epoch 4000. According to the graph, right after the start of fine-tuning around epoch 3000, the loss value temporally increases. In contrast, at epoch 4000, the loss value decreases at the same value as before fine-tuning. Moreover, the result of E7 is more accurate than that of E8. Even by fine-tuning, the gap in lighting conditions may affect the performance of the CG-based method. However, it takes much cost to prepare RGB images with ground truth labels for fine-tuning. The number of epochs or RGB-based images required is dependent on the required accuracy. This trade-off should be investigated further.

## 5. Conclusions and Future Work

In this paper, we proposed a method for estimating the 6DoF pose of an object from a monocular RGB image. To improve the accuracy and be robust to domain shift, we used edge information, which is a common feature for both simulation and physical situations. In the evaluation, we tested various conditions and parameters. In the simulation environment, the proposed method outperforms the previous method [7]. In the physical environment, however, the proposed method is less accurate than the previous method due to domain shift. We mentioned some solutions to domain shift. For example, by fine-tuning with physical data, the proposed method marks better accuracy than the previous method. We concluded that the experimental results revealed the effectiveness of edge-based pose estimation for objects having many straight lines.

There are several future works to consider. Although we adopted Pose Interpreter Networks as a previous method, we should confirm the performance improvement when other CNN-based pose estimation methods are used as an inference model. Moreover, we should compare the performance between recently published methods and Pose Interpreter Networks with our proposed preprocessing. Next, as mentioned in Section 4.4, we need to fill more gaps between simulation and physical. We aim to achieve data augmentation with respect to edge distribution or learning-based line segment detection. Transfer learning, including fine-tuning, also needs to be investigated further. In addition, the target object should be broadened. Although this method presumes the existence of a designed 3D model that has enough straight ridgelines, the method should be able to deal with other 3D formats, such as point clouds made by a multi-view 3D reconstruction technique to expand the use case.

## Figures and Tables

**Figure 1 sensors-22-09610-f001:**
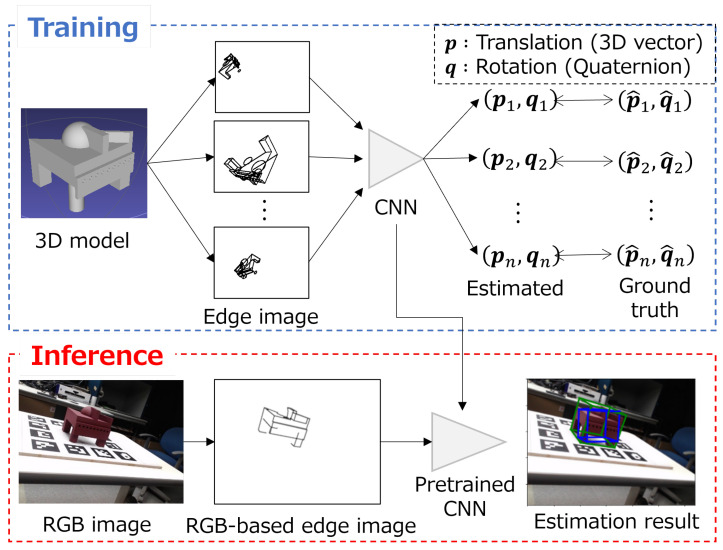
Overview of our method. Firstly, an edge image is randomly created by using the 3D model. The CNN model is trained with the generated dataset. At the inference stage, an RGB-based edge image is generated from an RGB image. Finally, the RGB-based edge image is input to pre-trained CNN and the pose is estimated.

**Figure 2 sensors-22-09610-f002:**
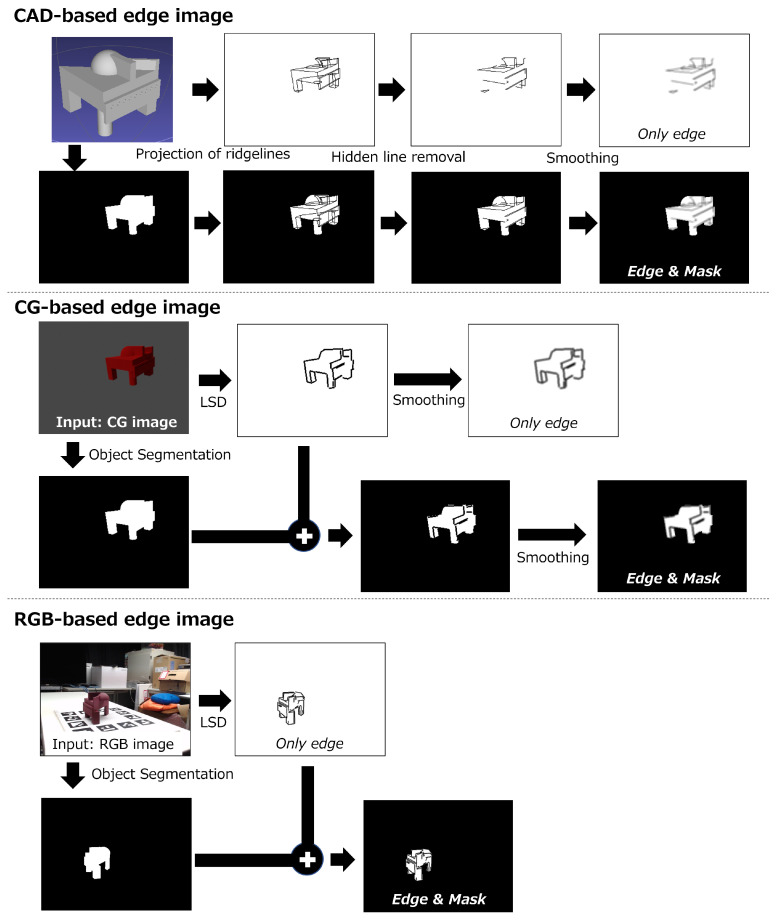
Process flow of edge image generation. Upper: CAD-based edge image. First, all ridgelines are projected. Then, hidden lines are removed. Finally, smoothing is adapted to each edge image. Middle: CG-based edge image. The 3D model is rendered by a renderer with certain environmental conditions. Then, line segments are detected and smoothing is adapted to each edge image. Lower: RGB-based edge image. The line segments are detected from RGB images. If a mask image is used, line segments are drawn onto the mask image.

**Figure 3 sensors-22-09610-f003:**
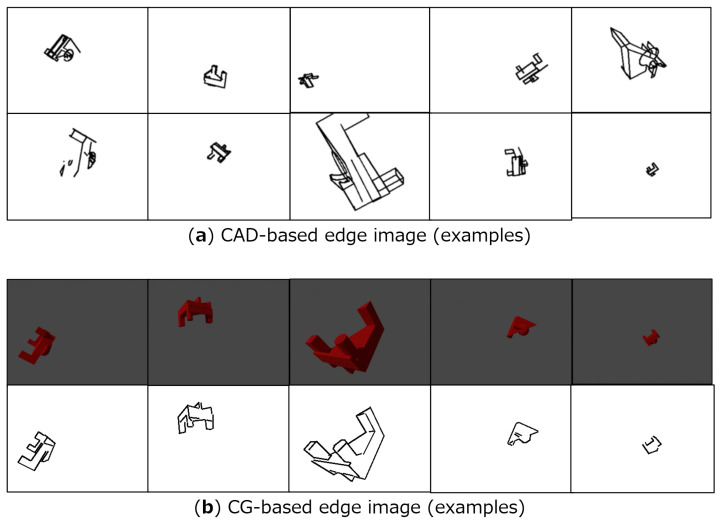
Examples of edge images created by simulation environment.

**Figure 4 sensors-22-09610-f004:**
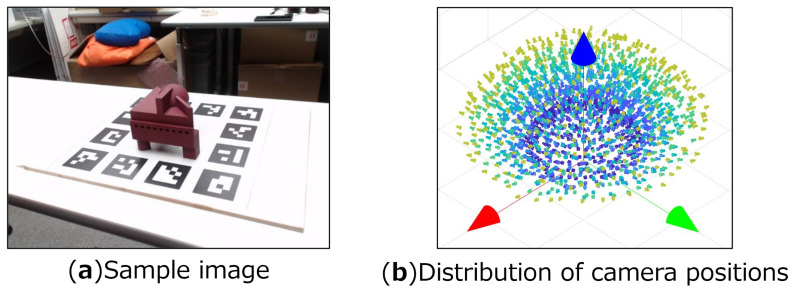
Physical dataset. (**a**) An example of the image. The 3D-printed object is placed on the AR marker. (**b**) The distribution of camera position. An object is placed on the origin of the coordinate.

**Figure 5 sensors-22-09610-f005:**
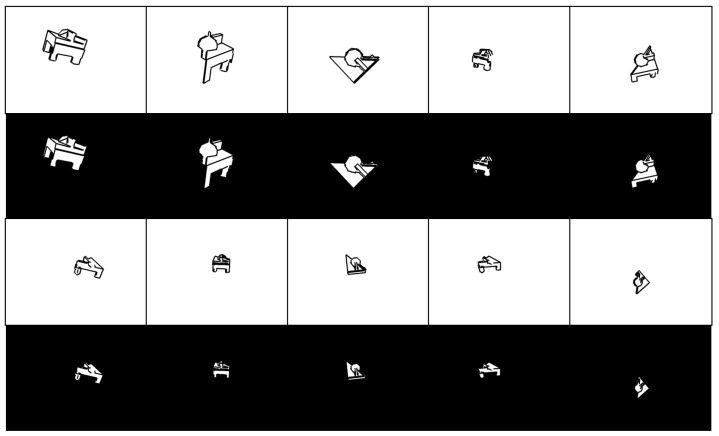
Examples of edge images created by physical environment. These images are width 2.

**Figure 6 sensors-22-09610-f006:**
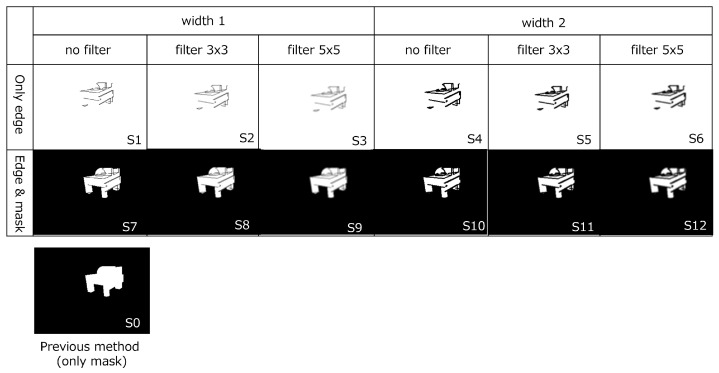
CAD-based edge images used at E1 and E2. *Width* means the width of the edge line (1 or 2) and *filter* means the size of the smoothing filter. Each condition has an *and mask* option mentioned in Figure 2. The previous method uses a silhouette image (S0). S1 to S12 denote the indices of the conditions proposed by us.

**Figure 7 sensors-22-09610-f007:**
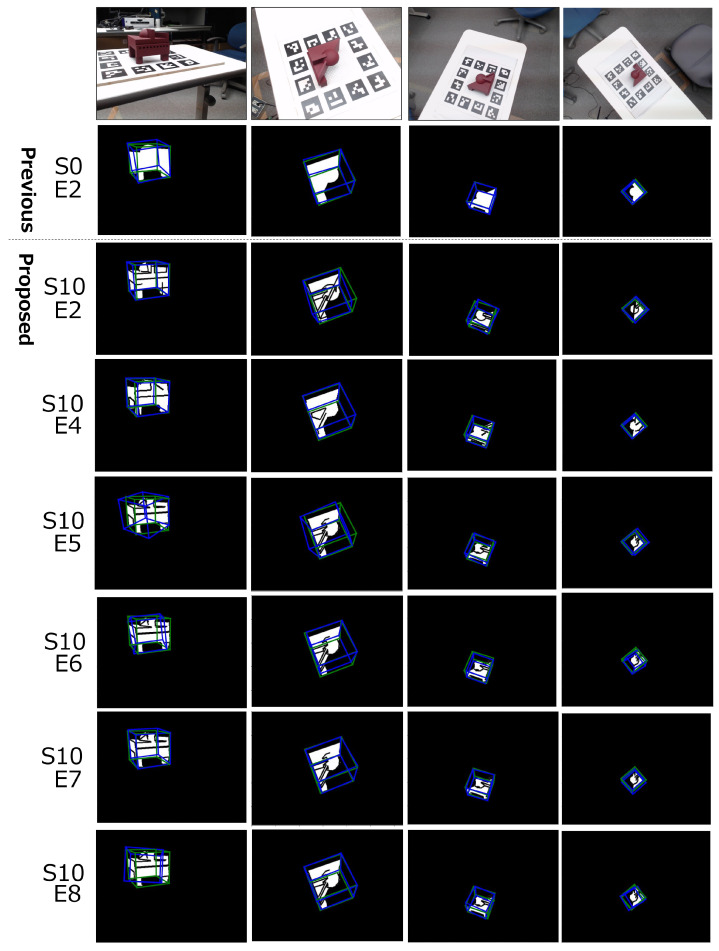
Examples of pose estimation results with the physical environment. The S0 row represents the result of the baseline method. Bounding boxes (BB) surrounding the object are calculated with pose information. Green BB is the ground truth pose and Blue BB is the estimated pose. Note that only the *edge and mask* result is represented because learning was not converged at several *only edge* conditions.

**Figure 8 sensors-22-09610-f008:**
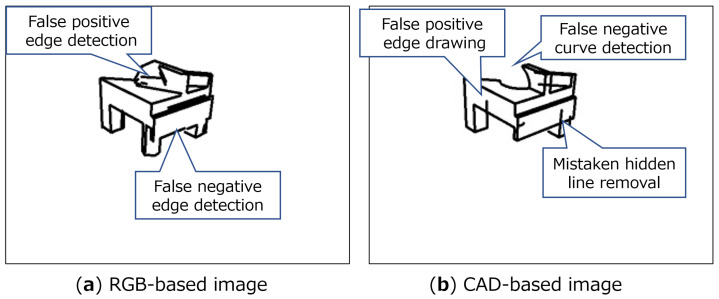
An example of the difference between RGB-based and CAD-based images.

**Figure 9 sensors-22-09610-f009:**
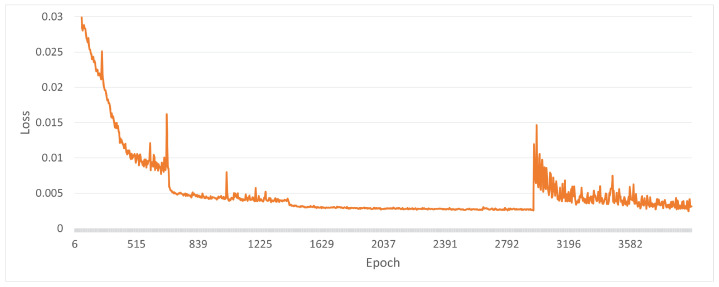
The example of a learning curve with fine-tuning (E7, S7). A vertical axis shows the loss value and a horizontal axis shows the number of epoch.

**Table 1 sensors-22-09610-t001:** Kinds of experiments described in this Section. There are three kinds of edge image generation methods: CAD-based, CG-based, and RGB-based. There are two types of pose generation: simulation and physical. *Training* means how to train the model. *Validation* means how to check valid training. *Inference* means how to estimate in a physical situation.

	Training	Validation	Inference
E1	CAD-based, Simulation	CAD-based, Simulation	
E2	CAD-based, Simulation		CAD-based, Physical
E3	CG-based, Simulation	CG-based, Simulation	
E4	CG-based, Simulation		CG-based, Physical
E5	CAD-based, Simulation		RGB-based, Physical
E6	CG-based, Simulation		RGB-based, Physical
E7	CAD-based, Simulation		RGB-based, Fine-tuning
E8	CG-based, Simulation		RGB-based, Fine-tuning

**Table 2 sensors-22-09610-t002:** Translation error. S0 is the previous method. S1~S12 are the proposed methods. ’s.d.’ means standard deviation. N/A means the training is not converged. ↓ means lower value is more accurate.

$e_{t}$ (Translation Error) (cm) ↓													
**Exp.**	**S0**	**S1**	**S2**	**S3**	**S4**	**S5**	**S6**	**S7**	**S8**	**S9**	**S10**	**S11**	**S12**
**Mask**	—							w/	w/	w/	w/	w/	w/
**Line width**	—	1	1	1	2	2	2	1	1	1	2	2	2
**Filter**	—	no	3 × 3	5 × 5	no	3 × 3	5 × 5	no	3 × 3	5 × 5	no	3 × 3	5 × 5
E1(mean)	1.70	44.48	N/A	N/A	12.69	N/A	44.07	5.45	1.13	1.73	1.31	55.93	22.31
E1(s.d.)	1.93	14.56	N/A	N/A	4.54	N/A	12.91	41.00	1.07	6.50	1.27	474.49	215.28
E2(mean)	1.56	48.88	N/A	N/A	13.14	N/A	52.40	2.06	2.39	7.25	1.42	4.20	4.09
E2(s.d.)	1.52	15.42	N/A	N/A	4.36	N/A	13.07	1.43	1.91	4.66	0.96	23.85	2.11
E3(mean)	1.70	12.71	N/A	N/A	8.98	N/A	N/A	4.37	1.15	14.25	6.64	10.57	1.35
E3(s.d.)	1.93	4.55	N/A	N/A	4.40	N/A	N/A	26.33	1.19	116.09	64.51	95.98	2.60
E4(mean)	1.56	12.71	N/A	N/A	10.86	N/A	N/A	1.00	1.20	3.31	1.24	3.00	6.29
E4(s.d.)	1.52	4.70	N/A	N/A	4.23	N/A	N/A	1.31	0.90	3.03	1.03	2.02	3.20
E5(mean)	1.56	49.50	N/A	N/A	18.11	N/A	53.47	2.55	2.29	6.91	3.03	4.26	4.95
E5(s.d.)	1.52	15.38	N/A	N/A	5.83	N/A	13.37	2.33	2.16	4.60	2.52	15.21	2.84
E6(mean)	1.56	14.95	N/A	N/A	11.85	N/A	N/A	6.70	7.69	8.07	4.21	6.40	9.34
E6(s.d.)	1.52	5.28	N/A	N/A	4.25	N/A	N/A	4.09	4.19	3.96	3.70	4.66	4.52
E7(mean)	2.56	57.80	N/A	N/A	26.57	N/A	32.95	1.64	1.47	2.66	2.03	8.15	6.37
E7(s.d.)	2.35	17.37	N/A	N/A	7.45	N/A	8.59	1.96	0.97	1.82	1.44	38.60	30.82
E8(mean)	2.56	55.90	N/A	N/A	44.01	N/A	N/A	4.39	1.93	5.95	2.94	4.68	2.05
E8(s.d.)	2.35	16.35	N/A	N/A	11.75	N/A	N/A	16.75	1.29	24.04	10.62	19.97	1.38

**Table 3 sensors-22-09610-t003:** Rotation error. S0 is the previous method. S1~S12 are the proposed methods. ’s.d.’ means standard deviation. N/A means the training is not converged. ↓ means lower value is more accurate.

$e_{r}$ (Rotation Error) (deg) ↓													
**Exp.**	**S0**	**S1**	**S2**	**S3**	**S4**	**S5**	**S6**	**S7**	**S8**	**S9**	**S10**	**S11**	**S12**
**Mask**	—							w/	w/	w/	w/	w/	w/
**Line width**	—	1	1	1	2	2	2	1	1	1	2	2	2
**Filter**	—	no	3 × 3	5 × 5	no	3 × 3	5 × 5	no	3 × 3	5 × 5	no	3 × 3	5 × 5
E1(mean)	13.83	121.95	N/A	N/A	67.71	N/A	118.48	8.76	7.18	8.30	7.83	10.75	9.20
E1(s.d.)	21.12	36.51	N/A	N/A	39.25	N/A	39.57	20.33	9.37	14.01	10.16	23.43	18.65
E2(mean)	11.22	124.12	N/A	N/A	61.60	N/A	85.07	9.93	14.33	89.79	7.38	8.14	9.82
E2(s.d.)	19.14	36.49	N/A	N/A	34.98	N/A	43.94	11.91	19.20	54.32	5.61	8.07	7.84
E3(mean)	13.83	65.21	N/A	N/A	85.73	N/A	N/A	9.64	7.44	10.24	9.78	8.85	7.76
E3(s.d.)	21.12	40.40	N/A	N/A	43.17	N/A	N/A	21.63	9.89	23.89	18.81	17.31	13.42
E4(mean)	11.22	59.28	N/A	N/A	88.78	N/A	N/A	5.44	6.88	27.99	6.33	17.20	53.48
E4(s.d.)	19.14	42.25	N/A	N/A	39.55	N/A	N/A	6.78	7.32	35.78	6.74	17.84	46.56
E5(mean)	11.22	135.71	N/A	N/A	61.89	N/A	85.78	16.90	17.95	84.18	16.51	15.18	20.50
E5(s.d.)	19.14	32.25	N/A	N/A	37.42	N/A	44.01	25.57	25.13	56.21	24.03	17.42	27.90
E6(mean)	11.22	89.90	N/A	N/A	93.54	N/A	N/A	61.57	71.43	77.36	27.30	48.64	83.38
E6(s.d.)	19.14	49.79	N/A	N/A	38.95	N/A	N/A	54.99	44.87	44.83	37.08	48.80	54.50
E7(mean)	21.82	110.76	N/A	N/A	45.47	N/A	82.24	10.80	10.56	16.81	9.22	9.46	11.32
E7(s.d.)	31.78	33.54	N/A	N/A	29.98	N/A	31.89	16.98	10.78	20.15	6.36	9.28	17.55
E8(mean)	21.82	104.83	N/A	N/A	119.93	N/A	N/A	16.75	12.61	17.35	13.72	13.42	14.40
E8(s.d.)	31.78	43.51	N/A	N/A	30.34	N/A	N/A	21.96	16.98	27.05	14.95	15.96	18.92

## Data Availability

Not applicable.

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
