# Peer review of "Object Pose Estimation Using Edge Images Synthesized from Shape Information"

_sensors, 2022, doi:10.3390/s22249610_

Round 1

Reviewer 1 Report

The paper proposes 6DoF pose estimation from a monocular image using edge information. The authors claimed their proposed method is better than the pose interpreter network. To verify their claim, they used some simulation data. Following are the comments regarding this paper.

1.       In the abstract, it is better to illustrate the effectiveness of the proposed paper in quantitative terms.

2.       Page 2, introduction section, It is unclear how the network trained on simulation data can overcome the problems of lighting conditions etc.

3.       Related work lack inclusion of recent papers (2021-2022) on 6DoF pose estimation like DSC-Poenet, ZebraPose etc.

4.       In the methodology section, page 4, the mathematical formulation of the proposed method is beneficial in understanding the technique and the free parameters available to tune for better performance.

5.       Line 103, page 6, Authors have not used the Oil change dataset because it contains few edges. So it is crucial to evaluate the performance of the proposed method on a different number of edges present in the image. What criteria will the proposed method be suitable for, and for How many edges?

6.       Page 6, line 218, other publically available datasets are available with many edges and lines. Please consider using them for the verification of your methodology.

7.       Tables 2 and 3 show the translation errors. Again, it is better to present the errors as mean and standard deviation.

8.       S1 to S12 may be explained in a table to show the difference.

9.       Results may be compared with other recently published methods as well.

Reviewer 2 Report

Author needs to provide the description of some critical parts of the study.

1-      Abstract could be more informative by providing results. I prefer to see some results in the abstract.

2-      The introduction needs to be more emphasized on the research work with a detailed explanation of the whole process considering past, present and future scope. How the present study gives more accurate results than previous studies? It needs to be strengthened in terms of recent research in this area with possible research gaps. It is strongly recommended to add a recent literature.

3-      There are too many basic and well know equations, authors is not proposing any new equation please reduce the equation or put it in annexure. Please avoid the detail derivation, please add these in the annexure.

4-      The author used Cad drawings for the training and validation. What happens if handmade drawings will be tested?

5-      Please avoid the basic details about the methodology in the introduction portion, the introduction portion, please use only the latest reference. Please reduce these sections.

6-      Please describe the important and novelty of the selected problem, data details. Please provide details about the selected problem. Please include the validation process on the unique problem.

7-      Explain Figures 9 more in detail with the results discussion.

8-      Author use different abbreviation at different places, which confused the reader, Please provide the list of the abbreviation, please use in the start.

9-      In the conclusion section, the limitations of this study, suggested improvements of this work, and future directions should be added

The author needs to address the abovementioned points for the betterment of the manuscript.

The paper is very well organize and well presented, therefore please accept this present work, with following minor suggestions 
